# Response Methodology Optimization and Artificial Neural Network Modeling for the Removal of Sulfamethoxazole Using an Ozone–Electrocoagulation Hybrid Process

**DOI:** 10.3390/molecules28135119

**Published:** 2023-06-29

**Authors:** Nguyen Trong Nghia, Bui Thi Kim Tuyen, Ngo Thi Quynh, Nguyen Thi Thu Thuy, Thi Nguyet Nguyen, Vinh Dinh Nguyen, Thi Kim Ngan Tran

**Affiliations:** 1Faculty of Chemical and Environmental Technology, Hung Yen University of Technology and Education, Khoai Chau District, Hung Yen 17817, Vietnam; nguyentrongnghia@utehy.edu.vn (N.T.N.); nguyennguyet@utehy.edu.vn (T.N.N.); 2Faculty of Chemistry, TNU-University of Sciences, Thai Nguyen City 25000, Vietnam; tuyenkimbui194@gmail.com (B.T.K.T.); dtz2157720203011@tnus.edu.vn (N.T.Q.); thuyntt@tnus.edu.vn (N.T.T.T.); 3Institute of Applied Technology and Sustainable Development, Nguyen Tat Thanh University, Ho Chi Minh City 70000, Vietnam

**Keywords:** ozone, electrocoagulation, sulfamethoxazole, optimization, removal efficiency

## Abstract

Removing antibiotics from water is critical to prevent the emergence and spread of antibiotic resistance, protect ecosystems, and maintain the effectiveness of these vital medications. The combination of ozone and electrocoagulation in wastewater treatment provides enhanced removal of contaminants, improved disinfection efficiency, and increased overall treatment effectiveness. In this work, the removal of sulfamethoxazole (SMX) from an aqueous solution using an ozone–electrocoagulation (O–EC) system was optimized and modeled. The experiments were designed according to the central composite design. The parameters, including current density, reaction time, pH, and ozone dose affecting the SMX removal efficiency of the OEC system, were optimized using a response surface methodology. The results show that the removal process was accurately predicted by the quadric model. The numerical optimization results show that the optimum conditions were a current density of 33.2 A/m^2^, a time of 37.8 min, pH of 8.4, and an ozone dose of 0.7 g/h. Under these conditions, the removal efficiency reached 99.65%. A three-layer artificial neural network (ANN) with logsig-purelin transfer functions was used to model the removal process. The data predicted by the ANN model matched well to the experimental data. The calculation of the relative importance showed that pH was the most influential factor, followed by current density, ozone dose, and time. The kinetics of the SMX removal process followed the first-order kinetic model with a rate constant of 0.12 (min^−1^). The removal mechanism involves various processes such as oxidation and reduction on the surface of electrodes, the reaction between ozone and ferrous ions, degradation of SMX molecules, formation of flocs, and adsorption of species on the flocs. The results obtained in this work indicate that the O–EC system is a potential approach for the removal of antibiotics from water.

## 1. Introduction

Antibiotics are increasingly used for the treatment of humans and animals; however, they are also persistent pollutants when discharged into the water system. The antibiotic resistance gene can develop with the prolonged presence of antibiotics in water, presenting a danger to humans and animals [1]. Sulfamethoxazole (SMX) is a sulfonamide antibiotic that is frequently used to treat livestock and humans. Consequently, SMX is one of the most popular antibiotics detected in wastewater. When SMX is present in the aqueous system for an extended period, it can cause the growth of antibiotic-resistant bacteria and the development genes that threaten the aquatic environment and human health [1,2,3]. Because of its stability (half-life period of 85–100 days) and polarity, SMX is challenging to remove from water via conventional processes such as biodegradation and adsorption [4,5]. Therefore, developing effective methods for removing SMX from water is necessary.

Ozonation is a promising and clean method for the treatment of various pollutants because of its simplicity and scalability. Ozone, with an oxidation potential of 2.07 V, is one of the strongest oxidizing agents and a common oxidant for treating organic pollutants. The use of ozone for removing antibiotics has been increasingly applied in recent decades. Although ozonation has many advantages over other methods, it still has limitations. Besides pollutant mineralization, various byproducts can form during the oxidation process. In [6], the authors found that the ozonation of SMX could occur through several pathways, leading to numerous byproducts that are also harmful to the environment. Therefore, the combination of ozonation with other methods, such as sonication [7,8], adsorption [3], and oxidation [9,10] has been investigated to enhance the SMX removal efficiency. However, these combinations have the drawback that is difficult to develop on a large scale or to remove byproducts completely.

Recently, the combination of ozonation and electrocoagulation for treating different pollutants in wastewater has been investigated and is a promising approach for removing persistent contaminants [11]. The main advantage of the ozone–electrocoagulation (O–EC) system is the combination of oxidation and adsorption of pollutants in the treatment [12]. When iron is used as the anode, Fe^2+^ ions are released into the solution and react with ozone through several reactions to form hydroxyl radicals (the most potent oxidant with a redox potential of 2.7). Radicals and ozone involve the oxidation of the pollutants; hence, the degradation efficiency is enhanced [13]. Then, the pollutants and their byproducts are adsorbed onto iron hydroxides (flocs) and settle down as slugs. The O–EC system has a high removal efficiency for numerous pollutants and requires reasonable energy consumption for operation [11,14]. This O–EC approach has been studied and used to treat various kinds of wastewater [12,13,15,16,17,18,19,20,21,22,23]. In [16], the O–EC system was used to treat industrial distillery effluent and could reduce 100% of color and chemical oxygen demand (COD). The O–EC system also showed high efficiency in removing the COD and color of cardboard factory wastewater, with 74.7 and 97.5%, respectively. Behin et al. [20] reported that the decolorization efficiency of synthetic wastewater containing Acid Brown 214 in an O–EC system 100% was higher than that of a zonation or electrocoagulation system. Song et al. [24] investigated the variables influencing the C.I. Reactive Black 5 removal efficiency via the O–EC system and found that the energy consumption was approximately 33 kWh/kg of COD removal, and the color-removal efficiency reached 94%. In [14], the energy consumption for treating wastewater containing Reactive Black 5 was about 115 kWh/m^3^. Accordingly, the O–EC approach is highly effective with emerging pollutants; however, the use of this system for the removal of SMX has been rarely reported.

The design of experiment (DOE) has been regularly utilized for studying and optimizing chemical and environmental processes because it shows the main advantages, such as the reduction in the number of experiments needed to be carried out and the assessment of the effects of parameters [25]. Significantly, the interaction effects between the factors on the process can be evaluated via mathematical models used in DOE [26,27,28]. Among various DOE methodologies, response surface methodology (RSM) is one of the most common techniques used for optimizing chemical and environmental processes [26,29]. RSM can establish the relationships between process variables and the response of the system and provide insight into the underlying physical or chemical mechanisms of the process. Moreover, RSM uses a relatively small number of experiments to build a model of the system, reducing the overall experimental effort required to optimize the process [30,31]. In particular, it can be used to predict the optimal conditions for the process, saving time and resources by avoiding trial-and-error experimentation [32]. The interacting impacts of the independent variables on the process can be assessed using RSM [33,34,35].

Modeling and simulating a process involves gathering information on how the process is likely to occur without conducting physical experiments [36]. The use of modeling is widely recognized in science and engineering for various processes. In recent years, there have been successful applications of powerful artificial intelligence (AI) prediction methods such as the adaptive neuro fuzzy inference system (ANFIS), random forest (RF), and artificial neural network (ANN) for modeling pollutant removal processes [36,37,38,39]. The ANN is modeled according to biological nervous processing and it is a popular choice for solving and modeling complex engineering systems because it is simple, robust, reliable, and nonlinear. The ANN model is capable of learning from experimental data to solve complex, nonlinear, multi-dimensional functional relationships without any preconceived notions about their nature [36,37,40]. By using a set of experimental data, ANN models can establish the nonlinear connection between independent and dependent variables [41,42]. Furthermore, ANN models are able to access the relative importance of each parameter in processes [43,44].

This work aims to optimize the parameters of the O–EC system for the removal of SMX using RSM. The effects and inter-effects of the crucial parameters such as current density, initial pH, time, and ozone dose on the SMX removal efficiency are investigated, and the optimum conditions for the removal of SMX are established. The removal process is modeled using an ANN, and the relative importance of the parameters are calculated. The kinetics and mechanism of the removal process are also studied and discussed.

## 2. Results and Discussion

### 2.1. RSM Optimization

#### 2.1.1. Model Evaluation

To select the appropriate model for predicting the SMX removal process using RSM, three models, including linear, two-factor interaction (2FI), and quadratic models, were evaluated, and the results are given in Table 1. Among the three models, the sequential *p*-value of the linear and quadratic models is less than 0.0001, implying that these models are significant. The lack of fit of the quadratic model is not significant while that of the two other models is significant.

According to the evaluation, a quadratic model was suggested for predicting the experimental data with the equation as the following:Y = β + β_1_A + β_2_B + β_3_C + β_4_D + β_5_AB + β_6_AC + β_7_AD + β_8_BC + β_9_BD + β_10_CD + β_11_A^2^ + β_12_B^2^ + β_13_C^2^ + β_14_D^2^,(1)
where Y is the response (the removal efficiency), β is the mean value of Y, β_i_ (i = 1–14) is the coefficient of each term in the equation.

#### 2.1.2. Model Analysis

The analysis of variance (ANOVA) was used to evaluate the statistical significance of factors, and the results are given in Table 2. The model F-value of 341.84 implies that the model is significant [33]. There is only a 0.01% chance that an F-value this large could occur due to noise. The determined coefficient (R^2^) is 0.991, confirming that 99.1% of the total variables were described by the model. The adjusted R^2^ of 0.989 indicates the high significance of the model, and the predicted R^2^ of 0.982 suggests that only 1.8% of the total variables are not explained by the model. These values are in good agreement with each other. The coefficient of variation (CV) is quite low (1.11), implying good precision and reliability of the experiments. Adequate precision measures the signal-to-noise ratio, which is desirable when higher than 4 [30]. In this work, the ratio of 65.725 indicates an adequate signal. This model can be used to navigate the design space.

The lack of fit F-value of 1.67 implies that the lack of fit is insignificant relative to the pure error. There is a 29.72% chance that a lack of fit F-value this large could occur due to noise. Non-significant lack of fit is good, signifying that the regression model is satisfactory for predicting the SMX removal process by the O–EC system.

The *p*-values of the model terms, including B, C, D, AB, AD, BC, BD, CD, A^2^, B^2^, C^2^, D^2^, are less than 0.0500, indicating that these terms are significant. The relationship between response and coded factors can be presented as the following.
Y = 93.51 + 8.28A + 3.51B + 2.54C + 4.12D − 0.95AB + 2.41AD + 1.46BC − 0.87BD + 1.52CD − 4.68A^2^ − 5.09B^2^ − 2.29C^2^ + 2.91D^2^
(2)

#### 2.1.3. Diagnostics

The normality of a data set can be diagnosed using a plot of the normal probability versus the residuals, and if the resulting plot is approximately linear, the data are normally distributed. As presented in Figure 1a, the straight-linear pattern of the plot indicates that the experimental data follow a normal distribution. The assumption of constant variance can be tested using the plot of the residuals vs. the predicted. The plot is used to detect non-linearity, unequal error variances, and outliers. As seen in Figure 1b, the plot is a random scatter, indicating that a transformation is not needed. According to the Box–Cox Plot, the current lambda is 1, and the best lambda is 1.62. The 95% confidence interval around this lambda confirms that a specific transformation is not recommended. The predicted vs. actual response plot shows a good agreement between these values, indicating that the RSM model can predict the removal process well.

#### 2.1.4. Effect and Interactive Effect of the Parameters on the Response

The individual and interactive effects of the independent variables on the response can be determined through three-dimensional (3D) response surface and contour plots. The effect of the current density, pH, time, and ozone dose and their combined effect on the SMX removal efficiency are presented in Figure 2. The effects of current density on the response can be clearly seen in Figure 2a–c. The removal efficiency increases with an increase in the current density from 10 to 30 A/m^2^, slightly changes in the range of 30–40 A/m^2^, and decreases when the current density is above 40 A/m^2^. This can be explained by the fact that the quantity of Fe(II) ions generated at the anode is proportional with the current density, according to Farafay’s law [45]. The mass of flocs increases with an increases of the concentration of Fe(II) ions. Additionally, Fe(II) ions react with ozone to form hydroxy radicals that play an important role in the oxidation of SMX [46]. Hence, the increase in current density enhances the removal efficiency. However, if the value of current density is higher than the limit value, the adverse effects as reduction of mass-transfer and heat generation can occur [12,18].

As presented in Figure 2a,d,e, the removal efficiency is remarkably dependent on the initial pH. The proper pH value for the removal process is in the range of 8–9, and if the pH is out of this range, the removal efficiency decreases. This can be ascribed to the ability to generate hydroxyl radicals, the formation of flocculants, and the existing forms of the pollutants species [22]. For the effect of time, the removal efficiency rises remarkably when the treatment time increases from 10 to 30 min, and then sightly changes after 30 min. The removal efficiency is also highly affected by the ozone dose. The percentage of SMX removed increases with the ozone dose from 0.1 to about 0.7 g/h; while the dose is above this value, the change in the removal efficiency is not profound, indicating the excess of ozone for degradation of SMX [15].

The interactive effects of the factors can be seen by examining how the surface of the graph changes as the levels of the input factors are changed. The interactive effects of the factors are also confirmed by the curvature pattern of these plots, suggesting that it is important to consider the inter-effects of the factors on the removal efficiency during treatment design.

#### 2.1.5. Numerical Optimization

In RSM, the optimization process is crucial part of the experimental study to find out the optimum process condition. It is performed by adjusting the independent variables to determine the suitable values that give the most desirable response. In this study, the maximum level of SMX removal efficiency was targeted with the importance of 5, while the independent factors were kept in the ranges with the importance of 3, as presented in Table 3.

According to the solution, the optimum conditions for the removal process are a current density of 33.2, pH of 8.4, time of 37.8 min, and ozone dose of 0.7 g/h and the predicted response is 99.75%. To validate the prediction of RSM, three experiments were carried out under optimum conditions. The removal efficiency obtained from experiments were 98.68, 99.27, and 99.56%. The errors between experimental and predicted values are below 2%, confirming that the developed model can accurately predict SMX removal via the O–EC system. In comparison to the conventional methods, such as adsorption on activated carbon [47] and biological degradation [48], the O–EC process provides a higher removal efficiency in a shorter time.

### 2.2. ANN Modeling

The experimental data were analyzed using a three-layer, feed-forward neural network structure consisting of input, hidden, and output layers, as illustrated in Figure 3. To train this ANN structure, the back-propagation algorithm was employed. To select a suitable tranfer function for the ANN model, three differentiable transfer functions including log-sigmoid (logsig), tan-sigmoid (tansig), and purelin were examined. These functions are presented as the following equations:(3)$$y=logsig\left(x\right)=\frac{1}{\left(1+\exp\left(-x\right)\right)}$$
(4)$$y=tansig\left(x\right)=\frac{2}{\left(1+exp(2*x\right)}-1$$
(5)$$y=purelin\left(x\right)=x$$

The selected functions were dependent on the values of MSE and R^2^. Table 4 shows the R^2^ and MSE values obtained from the ANN with different transfer functions in layer 1 and layer 2. The R^2^ values of the ANN structures with logsig-purelin and logsig-tansig are close to the unity, indicating that these structures can predict well the experimental data. The MSE values of these models are 2.646 and 7.806, implying that the data generated by the ANN models are close to the experimental values. With the lowest value of MSE, the logsig-purelin ANN structure was selected for modeling the removal process.

Figure 4 presents the correlation between the experimental data and the data predicted by the selected ANN model for training, testing, validation, and whole data sets. It can be seen that the correlation coefficients for all data sets are close to 1, confirming that the obtained ANN structure can be used for modeling the removal of SMX via the O–EC system.

An important feature of ANN is the ability to evaluate the relative importance of each variable to the output, according to the weights of the network [36]. Their relationship can be presented as the following equation:(6)$$I_{j}=\frac{\sum_{m=1}^{m=N_{h}}\left(\left(\frac{W_{jm}^{ih}}{\sum_{k=1}^{N_{i}}W_{km}^{ih}}\right)\times \left|W_{mn}^{ho}\right|\right)}{\sum_{k=1}^{k=N_{i}}\left\{\sum_{m=1}^{m=N_{h}}\left(\left(\frac{W_{jm}^{ih}}{\sum_{k=1}^{N_{i}}W_{km}^{ih}}\right)\times \left|W_{mn}^{ho}\right|\right)\right\}}$$
where *I_j_* is the relative importance of the *j*th factor on the removal efficiency, *W* is the connection weight, *N_i_* and *N_h_* are the numbers of input and hidden neurons; the superscripts ‘*i*’, ‘*h*’, and ‘*o*’ mean input, hidden, and output layers; and subscripts ‘*k*’, ‘*m*’, and ‘*n*’ refer to input, hidden, and output neurons [44,49]. The calculated relative importance of the four factors is presented in Figure 5.

The calculated relative importance of pH is 0.341 which is the highest, implying that pH is the most influential parameter. The second influential parameter is current density, with a relative importance of 0.236. Finally, ozone and time have a relative importance of 0.216 and 0.211, respectively. The ANN calculation shows that all four parameters significantly affect the SMX removal process

### 2.3. Kinetic Study

To identify the kinetics of the removal process, the experiments were carried out with an initial pH of 8.4, current density of 33.2, and ozone dose of 0.7 g/h. The TOC values of SMX solution were determined within 35 min. The experimental data were analyzed using the first-order kinetic model as the following equation:(7)$$\ln\frac{C_{t}}{C_{o}}=-kt$$
where C_o_ and C_t_ are the TOC values of SMX solution at the beginning and after treatment, t is the time of treatment, and k is the reaction rate constant. The results were presented in Figure 6.

The coefficient of determination of the linear regression is 0.996, implying that the kinetics of the removal process fits well to the first-order kinetic model. The calculated rate constant was 0.12 (min^−1^), revealing that the removal of SMX via the O–EC system is a fast process.

### 2.4. Possible Mechanism

During the removal of SMX via the O–EC system, various physical and chemical processes can occur, such as oxidation and reduction on the surface of electrodes, reaction between ozone and ferrous ions, oxidation of SMX, formation of flocs, and adsorption of species on the flocs. The possible mechanism of the SMX removal process via the O–EC system can be depicted in Figure 7.

At the anode, iron is oxidized to form ferrous ions, and at the cathode, water is reduced to form hydrogen gas and hydroxyl ions as the following:Fe → Fe^2+^ + 2e
2H_2_O + 2e → 2HO^−^ + H_2_

When ozone is introduced into the system, it reacts with ferrous ions to generate hydroxide radicals and ferric ions [22,23] as the following reactions:Fe^2+^ + O_3_ → (FeO)^2+^ + O_2_
FeO^2+^ + H_2_O → Fe^3+^ + HO^•^ + OH^−^
FeO^2+^ + 2H^+^ + Fe^2+^ → 2Fe^3+^ + H_2_O

The advantage of the O–EC system is the presence of both strongly oxidizing species ozone and radicals which can degrade SMX molecules to form various transformation products. The degradation of SXM in aqueous solution by ozone and radicals has been thoroughly investigated in the literature [2,3,6,7,8,9,10,50]. According to the results reported in the [6], the degradation of SMX by ozone is complicated with several pathways such as hydroxylation, oxidation, and S-N cleavage. In the hydroxylation pathway, hydroxyl groups can attack to the benzene ring or C=C bond in the isoxazole ring to form byproducts with a molecular weight higher than that of SMX. Additionally, the amino group on the benzene ring and methyl group on the isoxazole ring can be oxidized to form nitro and carboxyl groups, respectively. The authors also reported that the main degradation pathway is the cleavage of the sulphonamide bond, and the most abundant transformation product is 3-amino-5-methylisoxazole. Similar transformation products were also observed by Gao et al. [2].

## 3. Materials and Methods

### 3.1. Chemicals

All chemicals used in the present study were of analytical grade (purchased from Merck, Darmstadt, Germany) and used directly without further purification. Sulfamethoxazole (C_10_H_11_N_3_O_3_S, 99.8%) was purchased from the National Institute of Drug Quality Control, Viet Nam. All chemical solutions were prepared using double distilled water (DDW). The SMX working solution was prepared with a concentration of 100 mg/L and was used for study via dilution to the specified concentration.

### 3.2. Ozone–Electrocoagulation System

The ozone–electrocoagulation (O–EC) system applied for the removal of SMX is illustrated in Figure 8. Steel plates (99.5% Fe) with a length of 10 cm and a width of 5 cm were used as the electrodes, and the working surface area of each electrode was 60 cm^2^. A DC power with a current output range of 0–10 A and voltage output range of 0–40 V was employed as a power supply for the reaction system. Ozone produced from a commercial ozone generator (D1, Dr. Ozone) was continuously purged into the reactor via ozone diffusers. Each experiment was carried out with 500 mL of a solution containing SMX (50 mg/L) and NaCl (0.8%).

After a specific time interval, 2 mL of the solution was withdrawn, immediately mixed with 1 mL of 2% KI solution to quench ozone in the solution, and filtered for analysis. The SMX removal efficiency was calculated according to the following equation:(8)$$\mathrm{Removal}\;\mathrm{efficiency}\;(\%)=\frac{{TOC}_{o}-TOC}{{TOC}_{o}}*100$$
where TOC_o_ and TOC are the total organic carbon values at the beginning and after treatment. These values were determined using a multi N/C 3100 analyzer (Analytik Jena GmbH, Jena, Germany). The ozone doses were measured using the potassium iodide wet-chemistry method [51]. To ensure reproducibility, each experiment in this work was performed three times. The reported values were average values with accepted errors ≤ 3%.

### 3.3. Experimental Design

Before designing experiments, the screening experiments (they are not shown in this article) were carried out to evaluate the effects of the factors affecting the removal process and the range of each factor. After that, central composite design (CDC) was utilized to design the experimental parameters for SMX removal via the O–EC system. In this work, four factors, including current density, initial pH, time, and ozone dose, were the independent variables. Table 5 illustrates five levels for each variable: low (−1), central (0), high (+1), alpha (−α), and alpha (+α).

### 3.4. Structure of ANN Model

The ANN modeling for the removal process was carried out by analyzing the experimental data with an ANN model consisting of input, hidden, and output layers. The numbers of neurons were 4 (the independent variables) for the input layer and 1 (the removal efficiency) for the output layer. The data were categorized into three groups: training (70%), validation (15%), and testing (15%). The neural network tool of MATLAB software (R2018a) was used for modeling.

### 3.5. Statistics

The fitness between the experimental and the predicted data obtained from the models was evaluated based on the mean square error (MSE) and the coefficient of determination (R^2^):(9)$$MSE=\frac{1}{n}\sum_{i=1}^{n}{\left(y_{i,exp.}-y_{i,pre.}\right)}^{2}$$
(10)$$R^{2}=1-\frac{\sum_{i=1}^{n}{\left(y_{i,exp.}-y_{i,pre.}\right)}^{2}}{\sum_{i=1}^{n}{\left(y_{i,pre.}-y_{m}\right)}^{2}}$$
where n is the number of data points; y_i,exp._ and y_i,pre._ are the ith values (removal efficiency) obtained from experiments and ANN model, respectively; y_m_ is the mean value of y_i,exp._

## 4. Conclusions

The removal of SMX via the O–EC system was optimized by RSM, using central composite design. Four parameters affecting the SMX removal efficiency of the O–EC system, including current density, pH, reaction time, and ozone dose, were numerically optimized. The removal process could be presented by a quadric model equation. The optimum conditions were the current density of 33.2 A/m^2^, time of 37.8 min, pH of 8.4, and ozone dose of 0.7 g/h, and the removal efficiency reached 99.65%. A three-layer ANN with logsig-purelin transfer functions can accurately predict the removal process with an MSE of 2.646 and an R^2^ of 0.980. The relative importance of pH was the highest, followed by current density, time, and ozone dose. The kinetics of the SMX removal process followed the pseudo-first-order kinetic model with the rate constant of 0.12 (min^−1^). The removal mechanism involved various processes: (1) oxidation and reduction on the surface of electrodes; (2) reaction between ozone and ferrous ions; (3) degradation of SMX molecules; (4) formation of flocs and adsorption of species on the flocs. The results obtained in this work show that the O–EC system is an efficient approach for removing antibiotics from water.

## Figures and Tables

**Figure 1 molecules-28-05119-f001:**
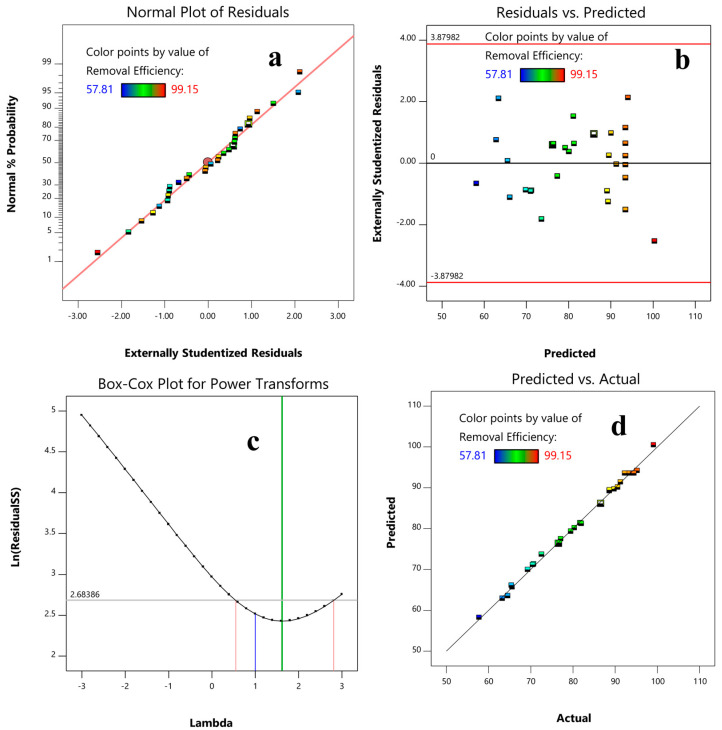
Normal plot of residuals (**a**), residual versus predicted plot (**b**), Box−Cox plot for power transform (**c**), and plot of predicted versus actual values of the removal efficiency (**d**).

**Figure 2 molecules-28-05119-f002:**
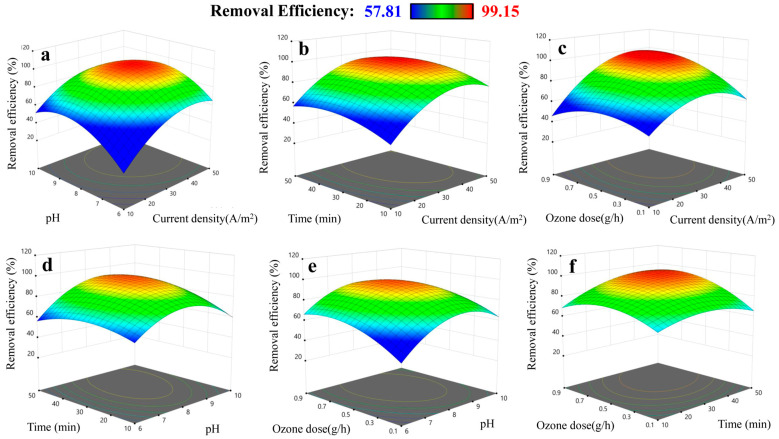
3D plots presenting effects of factors on the SMX removal efficiency: (**a**) pH and current density; (**b**) time and current density; (**c**) ozone dose and current density; (**d**) time and pH; (**e**) ozone dose and pH; (**f**) ozone dose and time.

**Figure 3 molecules-28-05119-f003:**
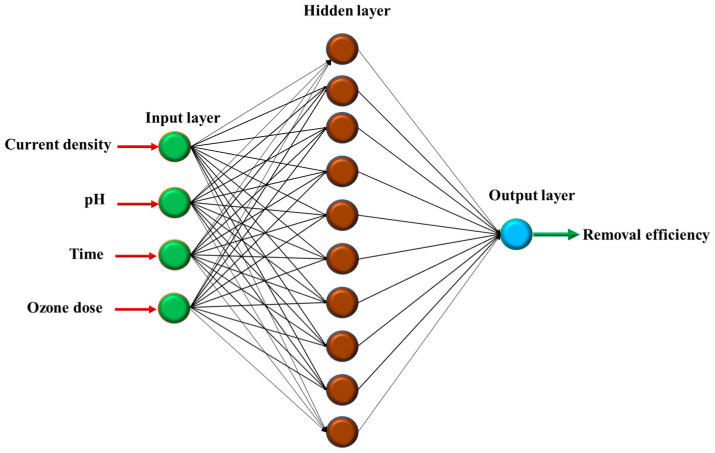
ANN structure used for modeling the SMX removal process by O-E system.

**Figure 4 molecules-28-05119-f004:**
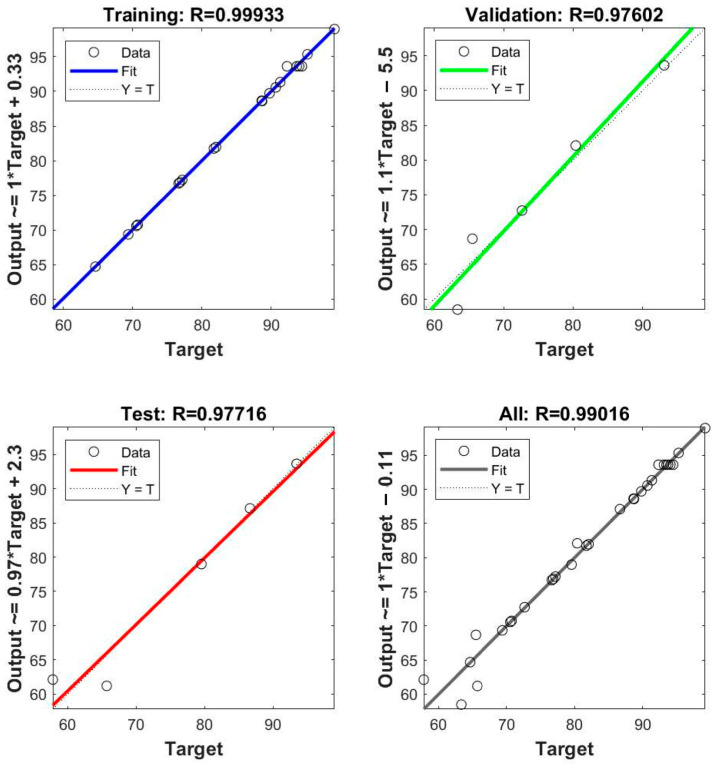
The final regression values of the ANN structure with logsig and purelin transfer function for trained, validated, tested, and whole network performance.

**Figure 5 molecules-28-05119-f005:**
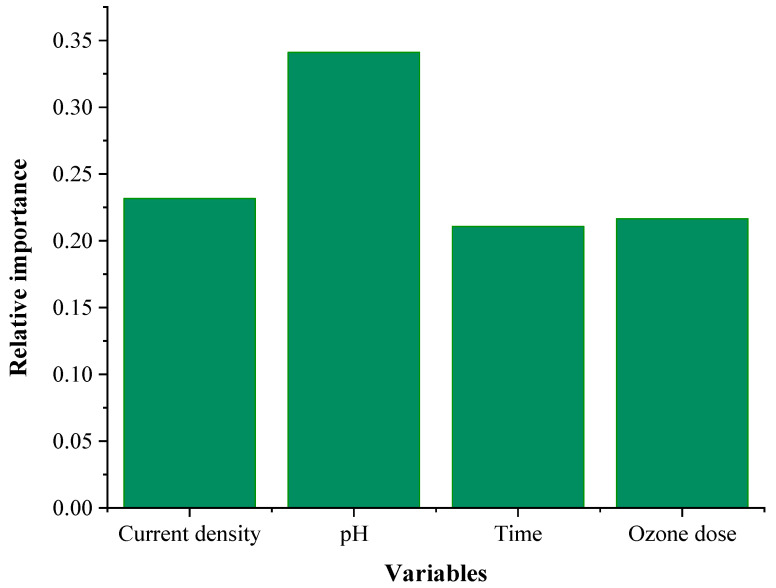
The relative importance of the variables calculated from the ANN.

**Figure 6 molecules-28-05119-f006:**
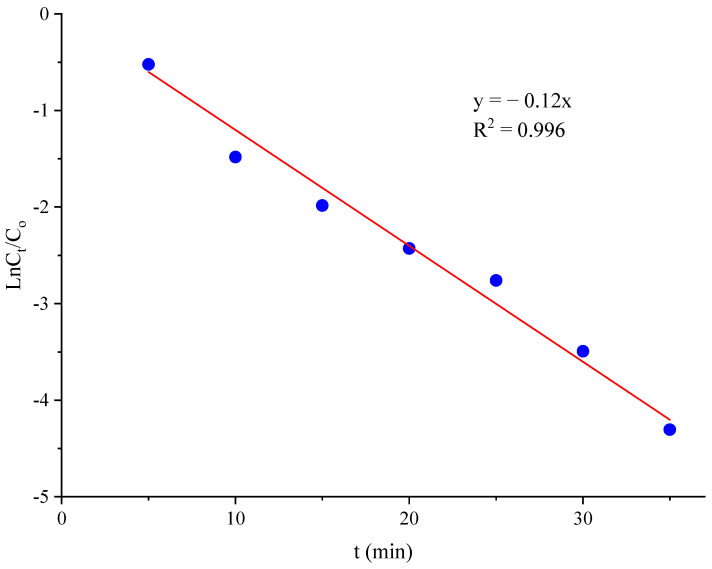
The first-order kinetic model for removal of SMX by O–EC system.

**Figure 7 molecules-28-05119-f007:**
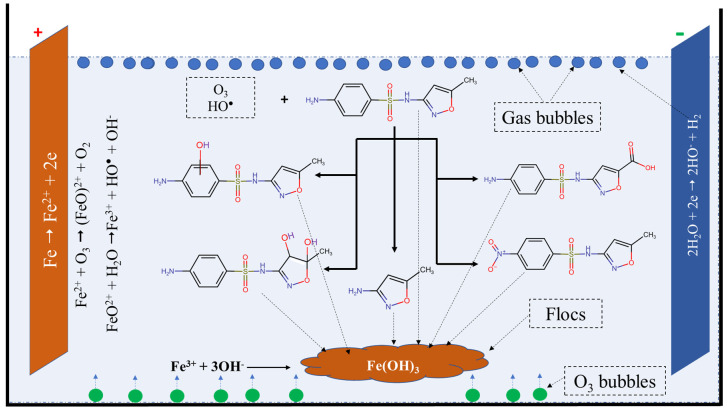
Possible mechanism for the removal of SMX by the O−EC system.

**Figure 8 molecules-28-05119-f008:**
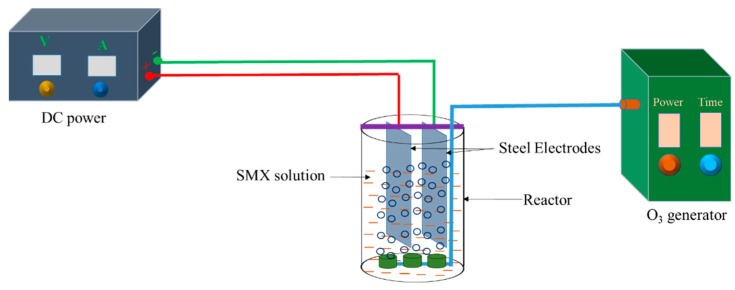
Schematic of O–EC system used for SMX removal.

**Table 1 molecules-28-05119-t001:** Model evaluation results.

Source	Sequential *p*-Value	Lack of Fit *p*-Value	Adjusted R^2^	Predicted R^2^
Linear	<0.0001	<0.0001	0.5734	0.5329
2FI	0.8169	<0.0001	0.5122	0.4673
Quadratic	<0.0001	0.2972	0.994	0.985

**Table 2 molecules-28-05119-t002:** ANOVA for quadratic model.

Source	Sum of Squares	df	Mean Square	F-Value	*p*-Value
Model	3948.36	14	282.03	341.84	<0.0001
A-Current density	1646.89	1	1646.89	1996.18	<0.0001
B-pH	295.19	1	295.19	357.80	<0.0001
C-Time	155.40	1	155.40	188.36	<0.0001
D-Ozone dose	406.64	1	406.64	492.89	<0.0001
AB	14.42	1	14.42	17.48	0.0008
AC	0.2426	1	0.2426	0.2940	0.5956
AD	92.79	1	92.79	112.46	<0.0001
BC	34.25	1	34.25	41.52	<0.0001
BD	12.20	1	12.20	14.78	0.0016
CD	36.88	1	36.88	44.70	<0.0001
A^2^	601.53	1	601.53	729.11	<0.0001
B^2^	711.82	1	711.82	862.78	<0.0001
C^2^	144.38	1	144.38	175.00	<0.0001
D^2^	232.35	1	232.35	281.63	<0.0001
Residual	12.38	15	0.8250		
Lack of Fit	9.53	10	0.9525	1.67	0.2972
Pure Error	2.85	5	0.5700		
Cor Total	3960.74	29			

**Table 3 molecules-28-05119-t003:** Criteria used for optimizing process.

Name	Goal	Lower Limit	Upper Limit	Lower Weight	Upper Weight	Importance
A: Current density	in range	20	40	1	1	3
B: pH	in range	7	9	1	1	3
C: Time	in range	20	40	1	1	3
D: Ozone dose	in range	0.3	0.7	1	1	3
Removal Efficiency	maximize	57.81	99.15	1	1	5

**Table 4 molecules-28-05119-t004:** MSE and R^2^ values obtained from ANN with different transfer functions.

Layer 1	Layer 2	R^2^	MSE
tansig	purelin	0.779	30.464
logsig	purelin	0.980	2.646
tansig	logsig	0.729	60.663
logsig	tansig	0.952	7.806
tansig	tansig	0.613	101.465
logsig	logsig	0.047	141.302
purelin	logsig	0.414	88.153
purelin	tansig	0.662	44.992
purelin	purelin	0.605	55.197

**Table 5 molecules-28-05119-t005:** Experimental factors and their five levels.

Independent Variables	Range				
	−α	−1	0	+1	+α
Current density (A/m^2^)	10	20	30	40	50
pH	6	7	8	9	10
Time (mm)	10	20	30	40	50
Ozone dose	0.1	0.3	0.5	0.7	0.9

## Data Availability

Data will be made available on request.

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
