# Peer review of "Response Methodology Optimization and Artificial Neural Network Modeling for the Removal of Sulfamethoxazole Using an Ozone–Electrocoagulation Hybrid Process"

_molecules, 2023, doi:10.3390/molecules28135119_

Round 1
Reviewer 1 Report
The paper is well-conceived and presented, and provides a comprehensive study on the optimization and modeling of the O-EC system for sulfamethoxazole removal. There are a few minor issues found that could be addressed to improve its quality.
Line 4: Introduce the abbreviation for sulfamethoxazole in abstract
lines 40-42. The references should be included to support the statement that SMX is challenging to remove from water using conventional processes
line 61. correct “reactions to form hydroxyl radials…”
line 67. Introduce COD abbreviation
-brief mention should be given about the energy requirements and efficiency of the proposed electrochemical method compared to other treatment options
Minor typographical errors and few grammar issues.
For example, in abstract:
"The data predicted by ANN model were matched well to the experimental data." should be "The data predicted by the ANN model matched well with the experimental data."
"The removal mechanism involved various processes such as oxidation..." should be "The removal mechanism involves various processes"
etc.
Author Response
Thank you for your valauble comments. Please see the attachment.

Reviewer 2 Report
What is the novelty and originality of this work? Which should be clarified in the introduction
The abstract section lacks background and a gap
The conclusion section is insufficient
The quality of figure 3 should be improved
One of the main criticisms is that a synthetic matrix was used, and the concentrations that were evaluated are not the typical ones found in bodies of water.
The electrocoagulation process is also not scientifically novel and its effectiveness has been proven in other publications. There is no comparison between this method and other more conventional options. So it is difficult to know if there would be any advantage in using this method.
Only one reference from the Molecules journal was added, therefore it does not present relevance with this journal
Therefore, I cannot recommend the submitted manuscript is published in Molecules in this way.
Author Response
Thank you for your valuable comments. Please see the attachment.

Reviewer 3 Report
The authors choose a very interesting topic. The removal of various substances, especially antibiotics, is of high interest. Not removing the antibiotics from waste waters can lead to resistant bacteriae. They made a quite Introduction section, but I would like to suggest giving more recent references than those cited (between 2008 and 2011). In Line 224, should it be Faraday's law instead of Farafay's? For conclusion I state this is a very decent paper. Therefore I would like to suggest to be accepted
Author Response

(The authors gave the same response as above.)

Round 2
Reviewer 2 Report
the authors responded to each of the comments in a satisfactory manner